# Morphometry of the Oculomotor Nerve in Duane’s Retraction Syndrome

**DOI:** 10.3390/jcm9061983

**Published:** 2020-06-24

**Authors:** Min Seok Kang, Hee Kyung Yang, Jounghan Kim, Jae Hyoung Kim, Jeong-Min Hwang

**Affiliations:** 1Department of Ophthalmology, Kim’s Eye Hospital, Seoul 07301, Korea; nietzsche@khu.ac.kr; 2Department of Ophthalmology, Seoul National University College of Medicine, Seoul National University Bundang Hospital, 82 Gumi-ro 173beon-gil, Bundang-gu, Seongnam-si, Gyeonggi-do 13620, Korea; nan282@snu.ac.kr (H.K.Y.); xyro_21@nate.com (J.K.); 3Department of Radiology, Seoul National University College of Medicine, Seoul National University Bundang Hospital, 82 Gumi-ro 173beon-gil, Bundang-gu, Seongnam-si, Gyeonggi-do 13620, Korea

**Keywords:** strabismus, Duane’s retraction syndrome, oculomotor nerve, extraocular muscle, high-resolution magnetic resonance imaging (MRI)

## Abstract

Objective: To investigate the morphometric characteristics of the oculomotor nerve and its association with horizontal rectus muscle volume in patients with Duane’s retraction syndrome (DRS) according to the presence of the abducens nerve. Methods: Fifty patients diagnosed with unilateral DRS were divided into two groups according to high-resolution magnetic resonance imaging (MRI) findings; DRS without an abducens nerve on the affected side (absent group, *n* = 41), and DRS with symmetric abducens nerves on both sides (present group, *n* = 9). Oculomotor nerve diameter was measured on high-resolution MRI in the middle of the cisternal space. The medial rectus muscle (MR) and lateral rectus muscle (LR) volumes were measured on T2-weighted coronal MRI of the orbit. Associations of oculomotor nerve diameter and horizontal rectus muscle volumes were performed according to the presence and absence of the abducens nerve. Results: Oculomotor nerve diameter on the affected side was thicker than that of the non-affected side in the absent group (*p* < 0.001), but not in the present group (*p* = 0.623). In the absent group, there was a positive correlation between oculomotor nerve diameter and MR volume (*r* = 0.779, *p* < 0.001), as well as the LR volume (*r* = 668, *p* = 0.023) of the affected eye. Conclusions: In DRS patients with an absent abducens nerve, the oculomotor nerve diameter was thicker in the affected eye compared to the non-affected eye. Oculomotor nerve diameter was associated with MR and LR volumes in the absent group. This study provides structural correlates of aberrant innervation of the oculomotor nerve in DRS patients.

## 1. Introduction

Duane’s retraction syndrome (DRS) is a rare form of congenital strabismus characterized by narrowing of the lid fissure during adduction, retraction of the eyeball, abduction limitation, varying degrees of adduction limitation, and displacement of the affected eye upwards and/or downwards during adduction [1,2,3]. Originally, DRS was regarded as a disorder related to extraocular muscle fibrosis, however, electromyographic studies have proven that innervational abnormality of ocular motor cranial nerves is the main etiology of DRS, namely, a congenital cranial dysinnervation disorder (CCDD) [4,5]. Pathologic case reports have shown that maldevelopment of the abducens nerve resulted in absent or markedly decreased action of the lateral rectus muscle (LR) on intended abduction, and branches of the oculomotor nerve were redirected to aberrrantly innvervate the LR, resulting in anomalous recruitment of the LR on intended adduction [6,7]. Depending on how many oculomotor nerve fibers normally bound for the medial rectus muscle (MR) get redirected in this manner, there may be essentially normal, slightly subnormal, or remarkably abnormal action of the MR [8].

There have been many studies regarding the presence and absence of the abducens nerve in patients with DRS using high resolution magnetic resonance imaging (MRI), in which the status of the abducens nerve is known to be closely related to the clinical manifestations of DRS [9,10,11,12,13,14,15,16,17]. However, structural features of the oculomotor nerve and its relationship with horizontal rectus muscle size and ocular motility has rarely been investated, which may be affected by aberrant innervation. Therefore, in this study, we aimed to demonstrate the morphologic characteristics of the oculomotor nerve and its association with ocular motility and horizontal rectus muscle volume in DRS patients. We divided patients according to the presence or absence of the abducens nerve, and the results were compared between groups [15].

## 2. Patients and Methods

A retrospective review of medical records was performed on 50 patients who had received a diagnosis of DRS between the years 2007 to 2016 at the Department of Ophthalmology, Seoul National University Bundang Hospital. All 50 patients had undergone high resolution MRI as previously described of which the ocular motor nerves on both sides were delineable in the cisternal space [15]. Ophthalmologic examination and MRI findings of the abducens nerve and oculomotor nerve at the brainstem level and orbit were noted. Diagnostic criteria of DRS included limited abduction and/or adduction, narrowing of the palpebral fissure on adduction, globe retraction, and upshoot and/or downshoot. The type of DRS was classified according to clinical observations; abduction limitation in type 1 DRS, adduction limitation in type 2 DRS, and both limited abduction and adduction in type 3 DRS. Patients were excluded if they did not have abnormal motility since birth and if the adduction or abduction deficit was related to other causative diseases, such as infantile esotropia or exotropia, acquired causes of oculomotor nerve palsy and abducens nerve palsy, internuclear ophthalmoplegia, synergistic divergence, or Möbius syndrome.

Ophthalmologic examinations included best corrected visual acuity (BCVA) in logarithm of the minimum angle of resolution (logMAR), ductions and versions, and binocular misalignment measured with the alternate prism and cover test or the Krimsky test. The degree of abduction and adduction were analyzed using image analysis of nine gaze photographs using the 3D Strabismus Photo Analyzer [18,19,20].

MRI was conducted using a 1.5 tesla system (Gyroscan Intera; Philips, Healthcare, Best, the Netherlands) or a 3 tesla system (Achieva; Philips, Healthcare, Best, the Netherlands) with a SENSE (SENSitivity Encoding) head coil. Thin-section T2-weighted imaging was performed to visualize the cisternal segment of the abducens nerve and oculomotor nerve in an axial plane through the brainstem, including the upper medulla oblongata, pons, and the midbrain. A 3-dimensional balanced turbo field echo sequence view of MRI was obtained as previously described [15]. The presence or absence of the abducens nerve was interpreted by one radiologist who was blinded to the clinical type of DRS in each patient. The diameter of the oculomotor nerve was measured at three locations, in the middle of the cisternal space and one section above and below it (Figure 1). Both sides of the oculomotor nerve were measured on the same level of the cisternal space by two independent ophthalmologists who were blinded to the patients’ clinical information. Finally, the average value of six measurements were taken for analysis. Evaluation of the abducens nerve was performed in a similar way described in a previous study [15]. The diameter of the abducens nerve was measured at three contiguous locations by two independent ophthalmologists the same way as the oculomotor nerve, and the average values were noted.

The right and left extraocular muscles were compared based on a side-by-side visual evaluation of their size and shape on the coronal T2-weighted images. The horizontal rectus muscle volume was measured on T2-weighted coronal images of the orbit. The sequence protocol of volume measuring was the same as that in our previous study [21]. MR and LR areas were measured in 3 to 5 contiguous coronal image planes, including the standard plane at the optic nerve-globe junction and the planes that were 2 mm and 4 mm anterior and/or posterior to the standard plane, as long as the border of each muscle was delineable. MR and LR volumes were defined as the average muscle area on all measured planes multiplied by 1 mm. The ratio of LR/MR volume was also noted to determine the structure–function relationship between horizontal rectus muscle volume and ocular motility.

Statistical analysis was performed using SPSS for Windows (Version 22.0, Statistical Package for the Social Sciences, SPSS Inc., Chicago, IL, USA). When variables showed a normal distribution, the paired *t*-test was used to compare both sides. Otherwise, the Wilcoxon signed rank test was used. Ocular motility and imaging parameters were compared between groups according to the presence (present group) and absence of the abducens nerve (absent group) by the independent *t*-test. *P* values < 0.05 were considered statistically significant. This study adhered to the Declaration of Helsinki and the protocol was approved by the Institutional Review Board of Seoul National University Bundang Hospital (IRB No. B-1304/200-106). Informed consent was not given, as patient records and information were anonymized and de-identified prior to analysis.

## 3. Results

### 3.1. Patient Characteristics

A total of 50 patients with DRS were included with a mean age of 7.9 ± 6.9 years (range, 1–38 years). Among them, 25 patients (50%) were men and 25 (50%) were women. In review of the patients’ past medical history, one patient had spina bifida and imperforate anus, one had intussusception, and another had otitis media, all of whom underwent surgery. One patient showed intellectual disability, and another one showed delayed growth.

All 50 patients had unilateral DRS (100%). Type 1 DRS was the most common type in 30 patients (60%), while 6 patients (12%) had type 2 DRS and 14 patients (28%) had type 3 DRS. The right eye was affected in 15 patients (30%) and the left eye in 35 patients (70%).

At the initial examination, the mean BCVA was 0.12 ± 0.14 logMAR in the affected eye and 0.11 ± 0.12 logMAR in the non-affected eye, which was not significantly different between both eyes (*p* = 0.854). Among all patients, 13 patients (26%) showed orthotropia, 19 patients (38%) had exodeviation without vertical deviation, four (8%) had exodeviation with hypertropia, 11 (22%) had esodeviation without vertical deviation, and two (4%) had esodeviation with hypertropia. One patient (2%) showed hypertropia without any horizontal deviation.

Grossly, the extraocular muscles including the MR and LR appeared symmetrical and normal-sized in all patients on coronal T2-weighted MRI. The oculomotor nerve on both sides were intact in all 50 patients.

Of the 50 patients, the abducens nerve on the affected side was absent in 41 patients (82%); 30 patients had type 1 DRS and 11 patients had type 3 DRS. The abducens nerve was present in 9 patients (18%), 6 patients with type 2 DRS, and 3 patients with type 3 DRS.

### 3.2. Groups According to Clinical Classification

The mean diameter of the oculomotor nerve on the affected side (1.819 ± 0.184 mm) was thicker than that of the non-affected contralateral side (1.754 ± 0.179 mm) in type 1 DRS (*p* < 0.001, paired *t*-test). However, there was no significant difference in type 2 DRS (*p* = 0.334, Wilcoxon signed rank test) and type 3 DRS (*p* = 0.147, Wilcoxon signed rank test) (Table 1).

The MR volume was smaller in the affected eye than the non-affected eye in type 1 DRS (*p* = 0.006), while there was no significant difference in type 2 and type 3 DRS (*p* = 0.334, *p* = 0.147, respectively).

The LR volume was smaller in the affected eye in type 1 and type 3 DRS (*p* = 0.017, *p* = 0.026, respectively), but not in type 2 DRS (*p* = 0.116).

The mean degree of abduction was 8.83 ± 7.31° in type 1 DRS, 28.88 ± 9.73° in type 2 DRS and 10.18 ± 7.36° in type 3 DRS. The mean degree of adduction was 25.41 ± 10.09° in type 1 DRS, 16.08 ± 8.72° in type 2 DRS, and 16.52 ± 7.29° in type 3 DRS. There was no significant correlation between the degree of abduction/adduction and absolute horizontal rectus muscle volumes.

### 3.3. Groups According to the Presence of Abducens Nerve

Imaging parameters were compared between groups according to the presence or absence of the abducens nerve (Table 2 and Figure 2).

In the absent group, the diameter of the oculomotor nerve on the affected side was thicker than that of the non-affected side (*p* < 0.001, paired *t*-test). MR volume and LR volume were both smaller in the affected eye than the non-affected eye (*p* < 0.001, *p* = 0.006, respectively) (Figure 2).

In the present group, there were no significant differences between both eyes regarding oculomotor diameter (*p* = 0.623, Wilcoxon signed rank test), and MR and LR volumes (*p* = 0.214, *p* = 0.091, respectively) (Figure 2).

In comparison of the absent group and present group, the diameter of the oculomotor nerve in the affected eye was not significantly different between both groups (*p* = 0.877). Oculomotor nerve diameter in the non-affected eye also showed no significant difference between both groups (*p* = 0.613).

Absolute MR and LR volumes of the affected eye were both smaller in the absent group compared to the present group (*p* = 0.005, *p* = 0.017, respectively). In terms of the non-affected eye, there was no significant difference in MR volume, however, LR volume was smaller in the absent group compared to the present group (*p* = 0.023).

Correlations between oculomotor nerve diameter and LR and MR volumes were analyzed in both the affected and non-affected eyes. In the absent group, there was a positive correlation between oculomotor nerve diameter and MR (*r* = 0.779, *p* < 0.001) and LR (*r* = 0.668, *p* = 0.023) volumes in the affected eye (Figure 3), whereas the non-affected eye showed no significant correlation. In the present group, no significant associations were found between oculomotor nerve diameter and horizontal rectus muscle volumes in both the affected and non-affected eyes.

To determine the structure–function relationship between horizontal rectus muscle volume and ocular motility, we incorporated the ratio of LR/MR volume. As a result, there was a moderate negative correlation between the range of adduction and LR/MR ratio in patients without an abducens nerve (*r* = −0.420, *p* = 0.026). No significant relation was found between ocular motility and horizontal rectus muscle volumes in the present group.

## 4. Discussion

In this study, we determined the morphologic characteristics of the oculomotor nerve and its association with horizontal rectus muscle volume and ocular motility. We divided patients according to the presence or absence of the abducens nerve, and subgroup analysis revealed significant findings mainly in the absent group. The major findings that we found in the absent group of DRS patients is as follows: Firstly, oculomotor nerve diameter in the affected eye was thicker than that of the non-affected eye. Secondly, MR and LR volumes were both smaller in the affected eye compared to the non-affected eye. Thirdly, there was a positive correlation between oculomotor nerve diameter and MR and LR volumes in the affected eye. Finally, there was a negative correlation between the degree of adduction and the ratio of LR/MR volume.

The overall mean diameter of the oculomotor nerve of the affected eye (1.817 ± 0.177 mm) was thicker than that of the non-affected fellow eye (1.765 ± 0.179 mm) in all patients. The mean diameter of the oculomotor nerve diameter in adult controls have been reported to be 1.9 mm (range, 1.6–2.2) on high resolution MRI, which is in line with our results [9,22]. A previous study on DRS patients reported MRI findings, suggesting aberrant innervation of the LR by fibers of the oculomotor nerve [23]. The abducens and oculomotor nerves are closely associated as they pass through the cavernous sinus and particularly as they enter the orbit through the LR [24]. The inferior division of the oculomotor nerve is divided into several branches penetrating the inferior medial aspect of LR suggesting the presence of aberrant innervation as well [6,7,10,25]. Based on these studies and theories, the results of our study infers a compensatory thickening of the oculomotor nerve related to aberrant innervation of the affected eye. It should be noted that these findings refer only to the absent group and not the present group, which implies a more widespread regeneration of the oculomotor nerve to the LR in the absent group.

MR and LR volumes were both smaller in the affected eye than the non-affected eye in the absent group. Possible explanations for LR hypoplasia in the affected eye are the absence of normal innervation by the abducens nerve, and subsequently, the antagonistic actions of the MR against LR may have been reduced. On the other hand, MR and LR volumes of the affected eye in the present group were significantly larger than those in the absent group (MR: *p* = 0.005, LR: *p* = 0.017, Mann-Whitney *U* test). This supports the paradoxical dual innervation of LR, partial innervations by the abducens nerve combined with aberrant branches of the oculomotor nerve [26]. Similarly, MR volume may get larger due to increased antagonistic action of the LR.

There was a positive correlation between the diameter of oculomotor nerve and MR (*r* = 0.779, *p* < 0.001) and LR (*r* = 0.668, *p* = 0.023) volumes in the absent group. In other words, the larger the diameter of the oculomotor nerve, the greater the volume of MR and LR, which indicates normal innervation of the oculomotor nerve to the MR and dysinnervation to the LR in case of an absent abducens nerve. This association structurally demonstrates that DRS is not only a congenital anomaly of the abducens nerve but is involved with aberrant innervation of the oculomotor nerve. On the other hand, there was no significant correlation between the diameter of the oculomotor nerve and MR volume in the non-affected fellow eye. Similarly, in the present group, there was no significant correlation between the diameter of the oculomotor nerve and MR volume in the affected eye. In our previous study, the diameter of the trochlear nerve positively correlated with superior oblique muscle volume in the non-affected fellow eye of congenital superior oblique palsy as well as normal controls [27]. However, the correlation between oculomotor nerve diameter and MR volume in normal controls is not well known, which may be more complex as the oculomotor nerve is a thick bundle of axons innervating many muscles other than the MR. 

Regarding the range of extraocular movement in DRS patients, the degree of adduction/abduction did not correlate with the absolute quantitative volume of horizontal rectus muscles. Instead, there was a negative correlation between the relative ratio of LR/MR volume and adduction in the absent group. As mentioned above, fibers destined for the MR get redirected to the LR by aberrant innervation [8,12]. Therefore, this negative correlation is consistent with the notion that the amount of aberrant innervation to the LR is inversely proportional to the amount of innervation to the MR, exerting an antagonistic action during adduction.

Our study should be interpreted in light of several limitations. The retrospective nature and small number of patients in the present group is a major disadvantage. We also excluded subjects with bilateral DRS, thus, future studies should enroll more of these patients to compare imaging characteristics with unilateral DRS that could give further insight into the pathophysiology of DRS. Another drawback of our study is that the results only provide indirect evidence of aberrant innervation, which should be supported by direct visualization of pathology or by higher resolution MRI. Finally, the lack of fixation during MRI causes variable eyeball positions that may affect extraocular muscle sizes. Additionally, as the direction of the LR muscle is not perpendicular to the coronal scan, the inclination of its course should be adjusted for volumetry. However, it is difficult to uniformly correct the inclined angle with respect to the coronal plane. To overcome these limitations, the LR volume was measured as the average muscle area obtained by three contiguous sections behind the optic-nerve globe junction, which is located posterior to the globe and short enough to be minimally affected by gaze and inclination. It seems reasonable to use this method as long as the same criteria is applied to all subjects.

## 5. Conclusions

In conclusion, unilateral DRS patients without an abducens nerve had a thicker oculomotor nerve in the affected eye compared to that of the non-affected eye. In these patients, there was a positive correlation between oculomotor nerve diameter and horizontal rectus muscle volumes in the affected eye. Our results provide structural correlates of aberrant innervation of the oculomotor nerve in DRS patients.

## Figures and Tables

**Figure 1 jcm-09-01983-f001:**
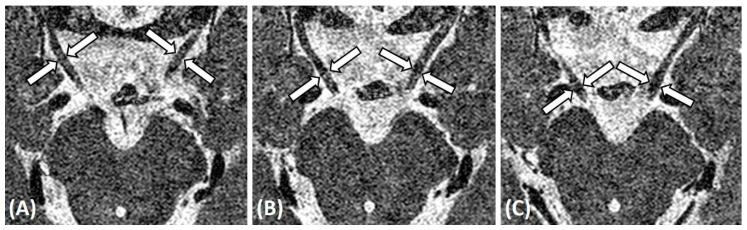
T2-weighted axial magnetic resonance imaging of the brain shows the cisternal segments of both oculomotor nerves (arrows) coursing in an anterior lateral direction at the level of the lower midbrain. The diameters of the oculomotor nerves were measured at three contiguous planes (**A**–**C**).

**Figure 2 jcm-09-01983-f002:**
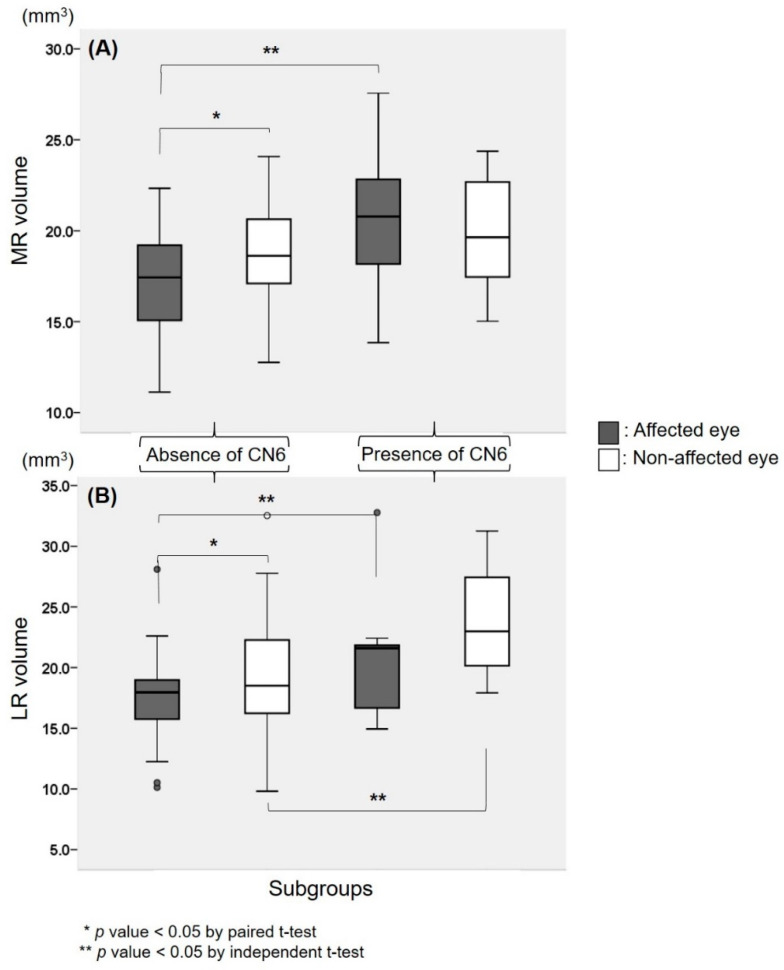
Medial rectus (MR) (**A**) and lateral rectus (LR) (**B**) muscle volumes in Duane’s retraction syndrome patients according to the presence and absence of the abducens nerve (CN6). MR and LR volumes were both smaller in the affected eye than the non-affected eye in the absent group (*p* = 0.001, *p* = 0.006, respectively). In the present group, there was no significant difference between the affected and non-affected eyes regarding MR and LR volumes (all *p* > 0.05). MR and LR volumes in the affected eye were both smaller in the absent group compared to patients in the present group (*p* = 0.005, *p* = 0.017, respectively).

**Figure 3 jcm-09-01983-f003:**
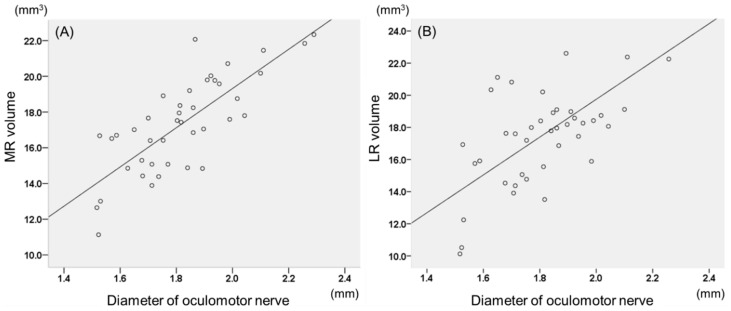
Correlations between oculomotor nerve diameter and horizontal rectus muscle volumes in patients without an abducens nerve in Duane’s retraction syndrome. (**A**) The medial rectus muscle (MR) volume of the affected eye showed a positive correlation with oculomotor nerve diameter (*r* = 0.779, *p* < 0.001). (**B**) The lateral rectus (LR) muscle volume of the affected eye also showed a positive correlation with oculomotor nerve diameter (*r* = 668, *p* = 0.023).

**Table 1 jcm-09-01983-t001:** Diameters of the oculomotor nerve (CN3) and abducens nerve (CN6) in Duane’s retraction syndrome patients according to their clinical classification.

	Cranial Nerve (CN)	Affected Eye (mm)	Non-Affected Eye (mm)	*p* Value *
Total (*N* = 50)	**CN3**	1.817 ± 0.177	1.765 ± 0.179	**<0.001**
	CN6	N/A	0.647 ± 0.067	–
Type 1 (*N* = 30)	**CN3**	1.819 ± 0.184	1.754 ± 0.179	**<0.001**
	CN6	N/A	0.644 ± 0.062	–
Type 2 (*N* = 6)	CN3	1.771 ± 0.106	1.742 ± 0.123	0.334
	CN6	0.666 ± 0.096	0.647 ± 0.088	0.345
Type 3 (*N* = 14)	CN3	1.831 ± 0.194	1.797 ± 0.205	0.147
	CN6	N/A	0.654 ± 0.072	–

Bold characters indicate that the differences are statistically significant; N/A, not applicable; * *p* value by paired *t*-test (*N* ≥ 30) and Wilcoxon signed rank test (*N* < 30).

**Table 2 jcm-09-01983-t002:** Diameters of the oculomotor nerve (CN3) and abducens nerve (CN6) in Duane’s retraction syndrome patients according to the presence or absence of the abducens nerve.

Groups	Cranial Nerve	Affected Eye (mm)	Non-Affected Eye (mm)	*p* Value *
Absent CN6 (*N* = 41)	**CN3**	1.818 ± 0.189	1.758 ± 0.189	**<0.001**
	CN6	N/A	0.645 ± 0.066	–
Present CN6 (*N* = 9)	CN3	1.808 ± 0.118	1.792 ± 0.129	0.623
	CN6	0.635 ± 0.089	0.656 ± 0.076	0.441

Bold characters indicate that the differences are statistically significant; N/A, not applicable. * *p* value by paired *t*-test (*N* ≥ 30) and Wilcoxon signed rank test (*N* < 30).

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
