# Peer review of "Morphometry of the Oculomotor Nerve in Duane’s Retraction Syndrome"

_jcm, 2020, doi:10.3390/jcm9061983_

Round 1
Reviewer 1 Report
The authors has presented here an interesting observation, but for the interest of the readers and also to give more information, it will be useful and interesting if the authors can provide
1) Details about other non-eye related medical issues in the study participants.
2) It will be interesting to know what is the Visual Acuity of these patients.
Author Response
The authors should address the following.
1. Details about other non-eye related medical issues in the study participants.
⇒ Yes, we added additional explanation in the results section as follows: page 3, section Results, first paragraph
“In review of the patients’ past medical history, one patient had spina bifida and imperforate anus, one had intussusception, and another had otitis media, all of whom underwent surgery. One patient showed intellectual disability, and another one showed delayed growth.”
2. It will be interesting to know what is the Visual Acuity of these patients.
⇒ Yes, we added additional explanation in the results section as follows: page 3, section Results, second paragraph
“The mean BCVA was 0.12±0.14 logMAR in the affected eye and 0.11±0.12 logMAR in the non-affected eye, which was not significantly different between both eyes.”
Thank you very much for your comment.

Reviewer 2 Report
Authors clarified structural relationship between the diameter of cranial nerves and the volume of extraocular muscles. This is very interesting and the novelty of this article is to be appreciated. Minor review: 1. It may be better to add sample MRI image to show which scans were used to measure diameters of the nerves. Did authors use different images to measure the diameter of oculomotor nerve and abducens nerve? It may be impossible to obtain a axial scan in which contains whole length of both nerves. 2. It is inappropriate to measure the diameter of the lateral rectus muscle on coronal images, because the lateral rectus muscle runs diagonally by approximately 45 degrees to the coronal scan. To use the term “volume”, the measured value should be corrected multiplying the coefficient. 3. Line 38 “Proved” may be “proven”.Author Response
1. It may be better to add sample MRI image to show which scans were used to measure diameters of the nerves. Did authors use different images to measure the diameter of oculomotor nerve and abducens nerve? It may be impossible to obtain a axial scan in which contains whole length of both nerves.
⇒ Yes, we added MR imaging in Figure 1 to show which scans were used to measure diameters of the cranial nerves. We used the same MR sequence to measure the diameters of oculomotor and abducens nerves. However, it was not measured with only one axial scan because the course of the two nerves are different. MR sequences were continuously obtained, the oculomotor nerve images throughout the level of midbrain and the abducens nerve images throughout the level of the pons. As described in figure 1, we measured 3 contiguous planes, twice each, and averaged the value for analysis.
“Figure 1. T2-weighted axial magnetic resonance imaging of the brain shows the cisternal segments of both oculomotor nerves (arrows) coursing in an anterior lateral direction at the level of the lower midbrain. The diameters of the oculomotor nerves were measured at three contiguous planes.”
2. It is inappropriate to measure the diameter of the lateral rectus muscle on coronal images, because the lateral rectus muscle runs diagonally by approximately 45 degrees to the coronal scan. To use the term “volume”, the measured value should be corrected multiplying the coefficient.
⇒ Thank you for your pertinent review. In case of the lateral rectus muscle, the volume can be changed depending on gaze and it is difficult to uniformly correct the inclined angle with respect to the coronal plane. To compensate for the shortcomings pointed out by the reviewer, the LR volume was measured as the average muscle area obtained by only 3 sections (total section thickness = 4 mm) behind the optic-nerve globe junction. A more extensive measurement of volume may lead to errors related to gaze direction and inclination of the LR. As the objective of our study was to compare the relative volumes of the affected vs non-affected, absent vs. present group, and LR/MR ratio, rather than to measure the absolute volume, it seems reasonable to use this method as long as the same criteria was applied to all subjects. We added this in the discussion as follows:
“Finally, the lack of fixation during MRI causes variable eyeball positions that may affect extraocular muscle sizes. Also, as the direction of the LR muscle is not perpendicular to the coronal scan, the inclination of its course should be adjusted for volumetry. However, it is difficult to uniformly correct the inclined angle with respect to the coronal plane. To overcome these limitations, the LR volume was measured as the average muscle area obtained by 3 contiguous sections behind the optic-nerve globe junction, which is located posterior to the globe and short enough to be minimally affected by gaze and inclination. It seems reasonable to use this method as long as the same criteria is applied to all subjects.”
3. Line 38 “Proved” may be “proven”.
⇒ Thank you for your careful review. Yes, we corrected it.
Thank you very much for your comments.

Round 2
Reviewer 1 Report
The authors has added the information in the manuscript. It can be accepted for publication.